# The future(s) of unpaid work: How susceptible do experts from different backgrounds think the domestic sphere is to automation?

**Vili Lehdonvirta**[1], **Lulu P. Shi**[1]*, **Ekaterina Hertog**[1,2], **Nobuko Nagase**[3], **Yuji Ohta**[3]

**1** Oxford Internet Institute, University of Oxford, Oxford, United Kingdom, **2** Ethics in AI Institute, University of Oxford, Oxford, United Kingdom, **3** Graduate School of Humanities and Sciences, Ochanomizu University, Tokyo, Japan

* lulu.shi@oii.ox.ac.uk

**Data Availability Statement:** All anonymised data from the Delphi survey are available from the UK

## Abstract

The future of work has become a prominent topic for research and policy debate. However, the debate has focused entirely on paid work, even though people in industrialized countries on average spend comparable amounts of time on unpaid work. The objectives of this study are therefore (1) to expand the future of work debate to unpaid domestic work and (2) to critique the main methodology used in previous studies. To these ends, we conducted a forecasting exercise in which 65 AI experts from the UK and Japan estimated how automatable are 17 housework and care work tasks. Unlike previous studies, we applied a sociological approach that considers how experts' diverse backgrounds might shape their estimates. On average our experts predicted that 39 percent of the time spent on a domestic task will be automatable within ten years. Japanese male experts were notably pessimistic about the potentials of domestic automation, a result we interpret through gender disparities in the Japanese household. Our contributions are providing the first quantitative estimates concerning the future of unpaid work and demonstrating how such predictions are socially contingent, with implications to forecasting methodology.

## Introduction

A 2013 working paper by Frey and Osborne [1] predicted that "about 47 percent of total US employment is at risk" from automation (2013: 1). The study spoke to emerging anxieties about "technological unemployment" in the wake of the financial crisis and was widely taken up in media and policy circles. Subsequent studies criticized the original study [2,3], reproduced it in other countries [4,5], and investigated automation's other impacts. A debate on the "future of work" began to command public attention, inform policy, and direct investments.

In this study, we broaden the future of work debate to unpaid work. Time use data reveals that the amount of time working-age adults spend on unpaid work in the domestic sphere is often comparable to the time they spend on paid work [6]. Unpaid domestic work comprises housework, such as cooking, cleaning, and washing clothes, as well as care work, such as childcare and caring for elderly relatives. In all societies, unpaid domestic work is

Data Service database (SN: 855342, DOI: 10.5255/
UKDA-SN-855342).

**Funding:** This research was supported by a UK-
Japan collaborative grant jointly awarded by UK
Research and Innovation (grant number ES/
T007265/1; PI Ekaterina Hertog) and by the
Research Institute of Science and Technology for
Society (RISTEX) of the Japan Science and
Technology Agency (grant number JPMJRX19H4;
PI Nobuko Nagase). This project also benefited
from funding from the European Research Council
(ERC) under the European Union's Horizon 2020
research and innovation programme under grant
agreement No 681546 (FAMSIZEMATTERS) and
grant agreement No 771736 (GENTIME). The
funders did not play any role in the study design,
data collection and analysis, decision to publish, or
preparation of the manuscript.

**Competing interests:** The authors have declared
that no competing interests exist.

disproportionately carried out by women [7]. Yet most of today's "future of work" discourse pertains to paid work only. What kind of futures are imagined for unpaid work? If robots will take our jobs, will they at least also take out the trash for us?

Drawing inspiration from the methodological approach used by Frey and Osborne [1,8] and its follow-up studies, we ask a panel of 65 "AI experts" to assess how susceptible to automation are 17 different types of domestic work. The experts predict that on average 39 percent of the time spent on a given task could be automated within ten years. Housework is on average seen as more automatable (44 percent) than care work (28 percent). Our results are broadly in line with previous research on paid work, but we for the first time present estimates specifically for domestic work tasks.

Where previous studies on the future of work have treated expert panels as a source of objective "ground truth", we apply a sociological approach to investigate also the socially contingent nature of such predictions. We recruit comparable numbers of female and male experts, and experts from different professional backgrounds. We examine how visions differ between these groups and offer potential explanations based on sociological literature and the experts' own written explanations. We also recruit experts from two different national contexts with diverging histories and public imaginaries of "AI" and automation: the United Kingdom and Japan. In the UK, the idea of technology replacing labour has been part of public consciousness at least since the Luddites. In Japan, robots are often viewed as a solution to problems of a rapidly aging society [9]. We examine differences in UK and Japanese experts' predictions in light of these discourses.

Our contributions are two-fold. First, we broaden the contemporary conversation around the future of work to include unpaid work in the domestic sphere, offering a modest corrective to the relative paucity of research on forms of work predominantly done by women [10,11]. Second, we demonstrate how expert opinion is socially contingent, and outline a more sociological and socio-technical approach to expert forecasting that acknowledges the contingency without dismissing forecasting as a methodology. For transparency, we publish individual experts' demographic backgrounds and automation predictions as open data in the UK Data Service.

Of the many possible impacts that automation can have on work, our study follows the stream of literature that focuses on the question of automation as a substitute to existing human labour. Thus a limitation of our study is that we do not consider automation's implications to issues such as task quality, skill development, or the generation of new tasks, which are addressed in other research.

## Background

In 2013, Oxford University researchers Frey and Osborne published a working paper titled, "The future of employment: How susceptible are jobs to computerisation?". They predicted that 47 percent of all jobs in the United States were at "high risk" of automation in perhaps "a decade or two" [1]. Although the prediction would not be published in a peer-reviewed journal until several years later [8], it immediately hit a nerve in the post-financial crisis era and was widely reported in Anglo-American media. With over 10,000 citations in Google Scholar, it is now far more cited than any other study on automation and employment.

The OECD, the UK Office for National Statistics (ONS), and various academics and corporate research institutes published replications and follow-ups to Frey and Osborne's study (e.g., [2,4,5]). "Future of work" grew into a major catchphrase among policy circles, think tanks, and academia. The shift in attention was followed by shifts in resource investment. Research funding agencies issued new funding calls; policy makers began to craft policies; investors reassessed their bets.

However, this "future of work" discourse has focused exclusively on paid work in the labour market. This is despite the fact that productive work in the labour market and reproductive work in the domestic sphere are just as vital for the functioning of societies. People in the UK aged 15 to 64 spend about 43 percent of all their work and study time on unpaid domestic work [6]. For statistical purposes, unpaid domestic work is defined using the "third party criterion" [12], which stipulates that domestic work is the sum of unpaid domestic activities that could be delegated to a paid worker or replaced by market goods.

Past technological transformations have implicated unpaid domestic work just as they have affected paid work. Time use research shows that time spent on many types of housework—such as cooking, cleaning, and laundry—decreased dramatically in the UK from 1920s to 2000s, as households adopted new appliances ([13]; but also because of changing practices; see [14]). In the near future, the adoption of new AI-powered technologies might be expected to result in further decreases. Without much fanfare, "service robots for domestic household tasks", largely comprised of robot vacuum cleaners and floor mops, have become the most widely produced and sold robots in the world [15]. The development of educational technologies for children gained renewed momentum from the pandemic. And as societies age, practitioners debate whether AI technologies could also replace more of the labour involved in caring for older adults at home [16].

Understanding these potential changes to unpaid work in the domestic sphere is clearly important for policy making and investment, yet no studies have attempted to quantify these potentials. This paucity of studies on unpaid work is consistent with a historical tendency of research to focus on employment to the exclusion of other forms of work, especially forms undertaken predominantly by women [10,11]. An emerging literature examines the impacts of automation on paid domestic work [17]. In this article we aim to use an approach similar to Frey and Osborne [1,8] and its follow-up studies to begin to produce quantitative predictions on the future of unpaid domestic work.

## A sociologically informed approach

The prediction methodology applied by Frey and Osborne [1,8] consisted of two steps. In the first step, a group of machine learning researchers "eyeballed" a list of 702 occupations and the job descriptions associated with each occupation in a U.S. occupational database; the researchers then "labelled" 33 of the occupations as "not automatable" and 37 as "fully automatable" [1]. In the second step, the authors applied a machine learning algorithm to generalise the labels to the remaining 632 occupations; the algorithm assigned an automation probability between 0 percent (not automatable) and 100 percent (fully automatable) to each occupation.

This is a typical workflow in machine learning research: people label some data points, the labelled data points are used to train an algorithm, and the algorithm is then used to classify further data points. The manual labels in this workflow are treated as ground truth—objective and accurate. Who the labellers are and how they arrived at their judgments is not considered important; the focus is on developing an algorithm that is able to reproduce their opinions accurately. Frey and Osborne [1,8] describe the manual labelling process only in passing, while spending over three pages describing the algorithm used to generalise the judgments.

When the data to be labelled are pictures of street signs, such an approach is not unreasonable. With suitable instruction, most people might produce similar labels. But when the data are complex constructs such as different types of work, then the subjective nature of the labels becomes obvious. The second aim of this article is therefore to acknowledge and address this issue in the dominant future of work methodology. This is especially important given that many follow-up studies have reused the labels that Frey and Osborne's [1,8] study produced.

Any weaknesses in the first step of the original study will now have been reproduced widely across the future of work literature.

A more sociological approach to knowledge production would recognize that material and social life structures people's consciousness, with the consequence that reality will be perceived differently by different experts as their material conditions differ [18,19]. Knowledge is materially, socially, and culturally "situated" [20]. The machine learning researchers that Frey and Osborne used as labellers were part of a particular "epistemic community" with its own culture of knowledge [21]. Drawing on their expertise was defensible on the basis that they were experts in many of the technologies used in task automation. But other professional communities might see the future differently. For instance, industry research and development (R&D) experts' views on the matter might be influenced by their experiences of applying these technologies in practice. Previous studies treat academic and industry experts as interchangeable.

Experts' judgments about automatability are also bound to be shaped by cultural discourses that to some extent define what is "reality" in any given society [18]. In the United Kingdom, John Maynard Keynes during the Great Depression in 1930s popularized the term "technological unemployment", which he defined as "unemployment due to our discovery of means of economising the use of labour outrunning the pace at which we can find new uses for labour" [22]. The term became popular again during the recessions of the late 1950s and early 1980s. In each case, commentators were concerned that unemployment was on the rise because investments into new technologies—called "labour-saving" or "automation" technologies— made it possible for firms to produce the same amount of output with fewer labourers [23].

Automation and technological unemployment entered public discourse in the UK again following the late 2000s financial crisis and the subsequent jobless recovery [24–26]. The crisis coincided with advances in machine learning techniques that were used to produce impressive public demonstrations of computers performing human-like feats. The term "artificial intelligence" or AI gained currency as popular shorthand for the latest generation of digital labour-saving technologies, whether or not they made use of machine learning as such [27].

In other countries, such as Japan, public discourses around technology and work have followed somewhat different trajectories [9]. Japan provides a good comparative case for the UK, because it is a similarly advanced industrialized country today but one with a rather different technological and economic history. Until the mid-20th century, Japan lagged considerably behind the UK and other rich European countries in technology adoption. Then in 1950s and 1960s, Japanese industries were transformed by massive private sector investment into technological innovation [28]. Thanks to co-determination policies, this technological transformation did not replace employment but instead increased employment security and wages, at least for male workers [29].

In recent decades, wages and employment security have eroded in Japan, too [30]. The global financial crisis caused a spike in unemployment; technological unemployment entered public discourse. In 2015, Nomura Research Institute collaborated with Frey and Osborne to replicate the famous study in the Japanese context, with the result that as much as 49 percent of the Japanese labour force was found to be at high risk of automation [4]. Yet survey evidence suggests that the general population in Japan still saw the prospects of technology replacing labour quite differently from the UK population. In a 2017 international survey conducted by Eurobarometer, 63 percent of British respondents agreed that "robots and AI steal peoples' jobs". In a matching 2019 national survey conducted by National Institute of Science and Technology Policy [31], only 23 percent of Japanese respondents agreed (the two surveys used slightly different response scales, but nevertheless provide roughly comparable data). And while 91 percent of UK respondents agreed that "robots and AI are technologies that require careful management" [32], only 67 of Japanese respondents agreed [31].

The two countries also have different histories of household automation. Similar to the UK, Japanese households from 1950s to 1970s rapidly adopted electric household durables, such as refrigerators and washing machines [33]. Yet surprisingly from a UK perspective, many other household technologies, including dishwashers and tumble dryers, never took on in Japan. Only 25 percent of households in Japan owned a dishwasher in 2014, compared to 41 percent in the UK in 2011 [34,35]. During pandemic school closures, only 3 percent of elementary school children in Japan had their own computer or tablet [36], compared to 33 percent in the UK [37].

Possible explanations include smaller homes and a generally lower female labour cost, resulting in Japanese households foregoing investment into bulky and expensive labour-saving technologies. According to Robertson [9], the Japanese robot industry also remains focused on "retro-tech", or "advanced technology in the service of traditionalism". In the domestic sphere this translates to household automation intended to lighten women's domestic burden without changing their role as the main homemaker. Against these diverging cultural discourses and material realities, Japanese and UK experts' visions of the future automatability of domestic tasks could well be expected to differ. Yet previous studies treat experts from different countries as interchangeable.

Finally, a more sociological approach to knowledge production should also have an interest in how the visions of marginalized people may differ from hegemonic discourses [18]. In all societies, unpaid domestic work is disproportionately carried out by women [7]. Feminist theory highlights how women's lived experiences give them access to different standpoints from which to view reality [19,20]. The domestic work disparity is especially stark in Japan, where the average working-age man's time spent on domestic work is only 18 percent of a working-age woman's; in the UK, the figure is 56 percent [38]. Women are thus likely to have accumulated much more experience of domestic work. Feminist scholarship on technology and work presents some good historical reasons for women to be sceptical of promises of technology alleviating this workload [39,40].

Other important determinants of subjective experience that might be expected to shape views on the automatability of domestic work include ethnicity and disability, including their various intersections. In this article we will not explore these dimensions further, but other scholarship has demonstrated how bias against such groups may be inadvertently built into AI applications (e.g., [41]).

## Methods

In this article we report results from an exercise carried out between September 2020 and June 2021 in which we invited experts to make judgments concerning the automatability of various types of work. The exercise is analogous to the first step in Frey and Osborne [1,8] and similar studies, except that the work is unpaid domestic work. We sought to improve on their method in several ways, namely by structuring the exercise as a forecasting exercise, recruiting experts from a variety of backgrounds, and being transparent about the expert group's composition. In other aspects we inherit the previous studies' methodological weaknesses, in particular the lack of a probabilistic sample formally generalizable to a population.

### Forecasting rather than labelling

Following established practice in machine learning research, Frey and Osborne understood their exercise as "labelling"; in social science research, the analogous method is "coding": categorizing data objects according to some scheme [42]. Coding tends to be carried out with greater attention paid to managing the subjective element; for instance, it is common to

employ several coders in parallel and to measure their consistency. However, estimating how automatable different types of work are is arguably neither a labelling nor a coding task—it is a forecasting task. The difference is that in forecasting, the participants are not asked to withhold their subjective judgment in favour of applying objective criteria; on the contrary, the whole method hinges on participants contributing their personal expertise [43].

Expert forecasting as a methodological approach is often used when statistical approaches to prediction are not sufficient or appropriate, such as when suitable past data are not available, the relevant theory remains unsettled, or discontinuities are expected [44,45]. Expert forecasting suffers from many biases and weaknesses, but among its advantages are that it can include private insider information, account for "soft" information that is difficult to quantify, and account for unusual events and discontinuities.

Participant selection is usually information-oriented: participants are selected on the basis of their assumed expertise or knowledge on the topic [46]. This can be formal expertise (e.g., academic) or domain knowledge accumulated by practitioners [45]. Homogeneous convenience samples tend to perform poorly compared to more diverse groups [46]. If participants are allowed to interact with each other, information exchange and group dynamics can lead to additional insights which may result in better accuracy than simply averaging across individual judgments [46]. However, interaction can also lead to counterproductive social dynamics such as conformity and domination by high-status opinion leaders, decreasing accuracy [43].

Probably the most widely used group forecasting method today is the Delphi method. The method originates from RAND Corporation research in the 1950s and 1960s [47]. It is defined by participant anonymity, iteration, controlled feedback, and statistical aggregation of responses [46]. In essence, participants are asked to make judgments in isolation, the results are fed back to the participants, and participants are asked to revise their judgments if desired. Thanks to these characteristics, the method can in theory generate some of the benefits of group interaction without many of its downsides, such as opinion leader domination [48]. RAND researchers applied the method to produce forecasts on things such as military outcomes, economic conditions, and future educational innovations. Today, the Delphi method is used in particular to produce long-range forecasts and forecasts concerning the effects of disruptive technological transformation, both of which are very difficult to forecast with purely statistical methods [43].

### Participant selection

Like Frey and Osborne [1,8], we targeted experts as our participants, except that instead of only machine learning researchers, we included a wider variety of "AI experts" in our sampling frame. Individuals were considered to be "AI experts" if they were identified as having significant expertise related to AI or related technologies such as machine learning, robotics, computer vision, and human-computer interaction, or the societal or business aspects of these technologies and their applications. We used a combination of desktop research (media, conference speaker lists, profile pages), our networks, and snowballing to identify a total of 285 experts based in the UK and Japan. We tried to identify comparable numbers of AI experts working in three different professional settings: academia, R&D (understood here as corporate research labs and start-ups), and business (e.g., venture capital and marketing). We also tried to identify comparable numbers of experts presenting as female and male. Since women are underrepresented in AI-related fields (e.g., [49]), we had to emphasise women in our search.

We reached out to the experts via email and LinkedIn messages, inviting them to participate in a Delphi forecasting exercise on the automation of domestic work. Communications with UK-based experts were conducted in English and with Japan-based experts in Japanese. In

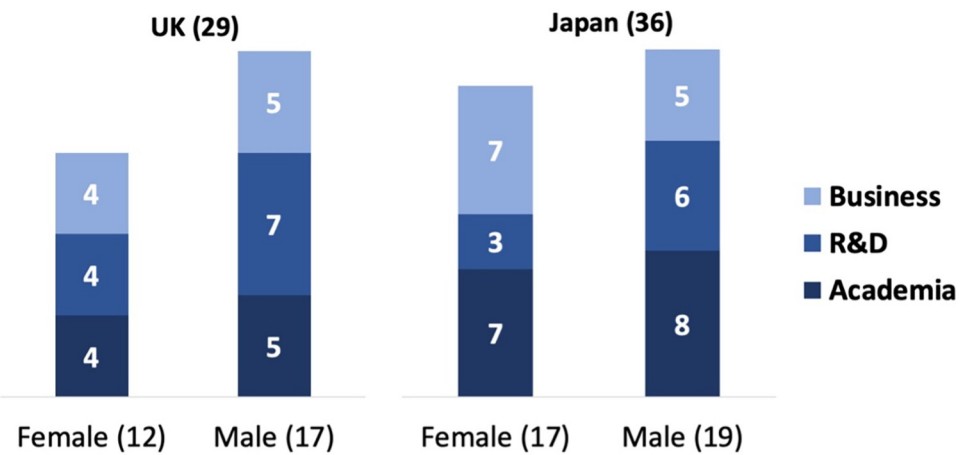

**Fig 1. Participating experts by country, gender, and professional background.**

appreciation of respondents' participation, we offered to share findings from our research with them. Participants in Japan were also offered a small monetary reward as is customary for expert interviews in Japan. Sixty-five experts agreed to participate (Fig 1). The response rate was higher in Japan than in the UK. Before participation each participant gave written informed consent. The study was approved by the Central University Research Ethics Committee at the University of Oxford, UK (approval number SOC_R2_001_C1A_21_52) and by the Humanities and Social Sciences Research Ethics Committee at Ochanomizu University, Japan (approval number 2020–86).

## Implementing the Delphi method

We guided our experts through a multi-round group forecasting exercise characteristic of the Delphi method [46]. The first round was aimed at defining the precise questions to be asked in the subsequent rounds. In the second round the questions were put to the experts. In the third round the experts had an opportunity to revise their answers based on other experts' answers. Finally, the answers were aggregated statistically.

In the first round the aim was to narrow down the precise questions. To this end we prepared a draft typology of unpaid domestic work tasks and draft questions to be asked about each task. The draft typology was constructed by combining the domestic work task typologies used in the 2014–2015 UK Time Use Survey (UKTUS, 93 tasks; see [50]) and the 2016 Survey on Time Use and Leisure Activities of Japan (STULA, 27 tasks; see [51]). The objective was to create a task typology that would 1) map to these two different but similar typologies and 2) make sense to experts who would be asked to assess the automatability of the tasks. We interviewed a small subset of the experts (three in the UK and seven in Japan) using parts of our draft typology and/or the draft questions as prompts and made revisions based on the feedback. Our final typology is presented in Table 1; it largely reflects established high-level categories in the time use research literature ([50,51]).

In the second round the aim was to elicit an initial estimate of the automatability of each task from each expert. We constructed an online survey instrument that presented each task along with a short description of the task and information on how much time women and men currently spent on the task (based on UKTUS and STULA). The instrument asked the experts, "[5/10] years from now, what percentage of the time that currently goes into this task can be automated?" (slider from 0 to 100, no default answer) and "[5/10] years from now, how

**Table 1. Typology of domestic work tasks.**

| Task | Category |
|---|---|
| Cooking | Housework |
| Dish washing | Housework |
| Household cleaning | Housework |
| Making and mending clothes | Housework |
| Laundry | Housework |
| Ironing and folding | Housework |
| Gardening | Housework |
| Pet care | Housework |
| Household and car maintenance | Housework |
| Grocery shopping | Housework |
| Shopping (other than groceries) | Housework |
| Using services | Housework |
| Physical childcare | Care work |
| Teaching a child | Care work |
| Interacting with a child | Care work |
| Escorting a child outside the home | Care work |
| Caring for an adult | Care work |

much will it cost for a household to use this automation for one year?". We instructed the experts to understand "automation" in "a very wide sense. . . referring to any use of 'AI' technology" (see the Appendix for the full definition and instructions given to experts). Since we only had 17 tasks compared to Frey and Osborne's 702 occupations, we asked each expert to manually evaluate every task. We then asked experts who had given the lowest and highest estimates for each task to supply written explanations for their answers. We received 63 explanations ranging from a single sentence to several paragraphs.

In the third round the aim was to give each expert the opportunity to revise their answers on the basis of what they learned from other experts. The survey instrument was amended to present summary statistics of the results of the first round for each task, along with the written explanations, anonymized and translated into English and Japanese. 57 of the 65 experts chose to revise at least some of their estimates, causing the estimates to converge somewhat; the average standard deviation of the 10-year automation estimate across all 17 tasks decreased from 27.92 to 23.89.

Finally, we aggregated the results by calculating the means for each task. In the following section we will present the results broken down by the experts' backgrounds, in this article focusing on the automatability percentage question. The cost question shows few discernible differences between expert groups owing to its categorical nature. Although the results are not drawn from a probabilistic sample, we conducted two-sided t-tests of statistical significance to identify differences that merit interpretation, labelled in the text as 'significant'. We offer interpretations for the differences based on previous literature and on a qualitative analysis of the written explanations, in which we coded the types of justifications given by experts from different backgrounds.

## Results

### Susceptibility of unpaid work to automation

Our expert panel estimated that on average 39 percent of the time that people currently spend on a given unpaid domestic work task could be automated within the next ten years; 27 percent could be automated within five years. The estimates varied significantly between tasks

**Table 2. Estimated automatability of domestic work tasks (% of time reduced).**

| Task | 5-year prediction (std.) | 10-year prediction (std.) |
|---|---|---|
| Grocery shopping | 45.25 (24.69) | 59.34 (23.40) |
| Shopping (other than groceries) | 39.17 (21.28) | 50.23 (22.71) |
| Using services | 36.86 (16.96) | 51.63 (20.42) |
| Household cleaning | 33.55 (19.81) | 46.18 (23.28) |
| Dish washing | 33.29 (24.91) | 46.89 (26.32) |
| Cooking | 32.48 (20.22) | 46.05 (23.06) |
| Ironing and folding | 30.68 (23.67) | 43.52 (25.72) |
| Laundry | 28.65 (21.89) | 42.95 (23.82) |
| Teaching a child | 27.40 (18.27) | 39.63 (22.19) |
| Gardening | 27.17 (20.20) | 39.63 (21.72) |
| Household and car maintenance | 24.89 (19.48) | 36.11 (21.55) |
| Caring for an adult | 23.77 (14.57) | 34.77 (17.56) |
| Making and mending clothes | 21.52 (23.87) | 29.06 (24.89) |
| Pet care | 21.09 (15.64) | 31.68 (18.63) |
| Interacting with a child | 14.72 (13.90) | 22.25 (17.67) |
| Escorting a child outside the home | 14.09 (11.66) | 23.55 (15.13) |
| Physical childcare | 12.32 (8.42) | 20.77 (12.90) |
| *Mean* | *27.47 (21.12)* | *39.07 (23.89)* |

(Table 2). The most automatable task was seen to be grocery shopping, of which 59 percent was considered automatable within ten years; the least automatable task was physical childcare, at 21 percent. In general, care work was predicted to be more difficult to automate, with an average estimate of 28 percent in ten years, while housework was seen as more readily automatable, at 44 percent.

These differences between task types are broadly speaking in line with previous literature on paid work. Economists have long held that "routine" work is more susceptible to automation than "nonroutine" work; complex problem-solving and communication tasks have been singled out as particularly resistant to automation [24]. Similarly, Frey and Osborne's experts predicted that occupations involving "assisting and caring for others", "persuasion", "negotiation", and "social perceptiveness" would be at a relatively low risk of computerisation [1], while Josten and Lordan [52] recently found that jobs involving non-linear abstract thinking and a high degree of social interaction were more difficult to automate. In their written explanations some of our experts likewise highlighted technical constraints: "we are still focusing on automation in this area, so it is unlikely that the technology will improve dramatically in the next five years", one wrote. This suggests that technology may not provide any rapid solutions to the issue of caring for an aging population at home.

However, the bulk of the experts' written explanations for why care work was difficult to automate actually focused on constraints that were not technical in nature, such as the social acceptability of delegating childcare to machines, its developmental impacts on the child, and its privacy implications. "I consider this to be the very last thing society will ever accept as suitable for automated services", wrote one. The opinion of the child was also mentioned as a factor (though not the opinion of elderly adults). Thus, even though we had followed previous "future of work" literature in explicitly instructing the experts to consider only the technical feasibility of automating various tasks (see the Appendix), we found that many experts appeared unable or unwilling to evaluate technical possibilities in isolation from societal factors that they saw as shaping the technology. As one expert explained:

I know you said not to consider whether people would want the tasks automating... But... what sort of products/tools/whatever the market will bring to people [is] based on what they are likely and reasonably able to use... as well as what is technically possible.

Similarly, experts noted how household budgets shaped what kinds of technologies were developed and marketed. One wrote:

[M]ost manual tasks can be automated as long as we can design the appropriate environment around them. So, the 'logically possible' degree of automation is high, the [real bottleneck is the] cost of that automation.

## Diverging visions between countries and genders

Though experts generally speaking agreed on which tasks were more and less susceptible to automation, individual experts' predictions varied significantly, as reflected in the standard deviations reported in Table 2. The most pessimistic expert predicted that on average across all tasks, only 9 percent of the time spent could be automated within the next ten years. For the most optimistic expert the average prediction was 72 percent. As expected, this variation was not random, but associated to some extent with the experts' backgrounds.

UK-based experts on average believed that automation could replace more household labour than their Japanese counterparts did: their average predictions were 42 percent vs. 36 percent in Japan in ten years, and 32 percent vs. 24 percent in five years. These differences are consistent with the differences in discourses around automation in the two countries as discussed in the background section; in the UK, technology is associated more with labour replacement. We did not detect any obvious qualitative differences in how our UK and Japanese experts explained their predictions, however, and we offer this only as a tentative interpretation.

Over the entire sample, female and male experts' predictions did not on average differ significantly from each other. But examining gender differences by country reveals a different picture (Table 3). In the UK, male experts were significantly more optimistic about technological potentials than were female experts, which is in line with the finding that men tend to be more optimistic about technology in general [53]. Yet in Japan, the situation was opposite: male experts were less optimistic than females.

Why were male experts in Japan comparatively pessimistic about the prospects of domestic automation? Possible interpretations could be sought from Japan's especially stark gender disparities and how that influences' experts' perceptions [54]. It would not be uncommon for Japanese men in expert positions to have almost no personal experience of major domestic work tasks, which they delegate instead to their wives. In the STULA 2016 survey only 52 percent of Japanese men aged 20–59 reported doing some domestic work (compared to 88 percent in the UK according to UKTUS 2014–15). Indeed, in their written explanations our Japanese male experts justified their pessimism in part by referring to what they saw as low demand for domestic automation considering its high cost. In contrast, Japanese women in expert positions have likely had to struggle significantly to balance work and family to reach such

**Table 3. Female and male experts' average predictions by country (% of time reduced).**

|  | UK | Japan |
|---|---|---|
| Women, 5 years (std.) | 28.11 (22.99) | 26.25 (18.53) |
| Men, 5 years (std.) | 34.08 (23.72) | 22.23 (17.82) |
| Women, 10 years (std.) | 38.04 (25.34) | 37.50 (22.23) |
| Men, 10 years (std.) | 45.57 (25.50) | 35.33 (21.76) |

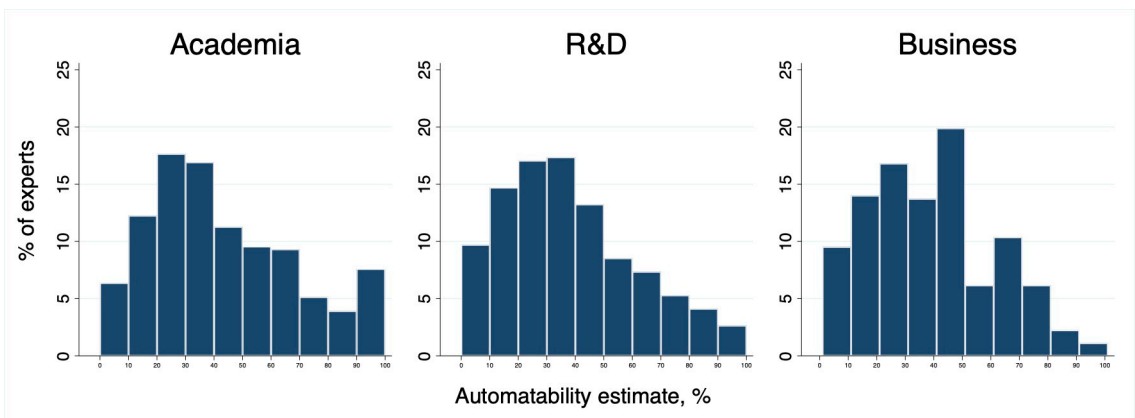

**Fig 2. Distribution of estimates by experts' professional background (10-year predictions).**

positions; as a result, they may see greater demand for domestic automation technologies even at a high cost. One Japanese female expert justified her relative optimism as follows:

[Already] in the case of household appliances that are used every day and are effective for domestic tasks, such as vacuum cleaners and rice cookers, people tend to buy high-priced and high-performance appliances.

### Varieties of AI expertise

Finally, we compared predictions of experts from different professional backgrounds. Though we are limited to three very high-level categories, some differences are apparent: on average, experts from academic and business backgrounds were more optimistic about the potentials of domestic automation than were experts from R&D (40, 42, and 35 percent in ten years, respectively). The academics' estimate has the highest standard deviation of the three, suggesting more disagreement. Indeed, Fig 2 reveals that while the majority of the academics' estimates are between 0 and 40 percent, there is a minority that exceeds 90 percent. This can be interpreted as reflecting the heterogeneous nature of academic AI expertise, such as pessimistic social scientists and optimistic machine learning research group leaders, for instance.

R&D experts' estimates fall smoothly on a bell curve. Written explanations from R&D experts were most likely to contain confident technical predictions. Experts from a business background had disproportionately chosen 50 percent as their prediction, perhaps reflecting doubt or indecision. Their written explanations often referred to demand considerations. The shapes of these distributions were very similar in both the UK and Japan, except that in Japan the R&D experts in particular were significantly less optimistic across the board than in the UK. It thus appears that the future of unpaid work somewhat depends on which type of expertise we draw on.

### Discussion and conclusions

In this article we extended the debate on the "future of work" to unpaid work by applying, for the first time, a Frey and Osborne [1,8] style expert panel method to produce predictions on the susceptibility of domestic work tasks to automation. This way we sought to offer a modest corrective to the lack of attention afforded to unpaid domestic work in analyses of work and labour [10,11]. This matters today, because immense resources, private and public, are being directed towards trying to manage the future of work.

Where Frey and Osborne [1,8] predicted that 47 percent of U.S. jobs were at high risk of automation, we predict that on average 39 percent of domestic work tasks could be automated

within ten years. This is an average figure across a typology of 17 housework and care work tasks; it does not take into account the different weights that these tasks in practice have in people's daily lives, but it does give an indication of the magnitude of possible labour savings at home. Neither Frey and Osborne's prediction nor our figure are intended as prognoses of what will inevitably happen, but rather as estimates of where the technical frontiers lie. If it is true that robots are taking our jobs, then it appears that they are also capable of taking out the trash for us.

Considering that people currently spend almost similar amounts of time on unpaid work as they do on paid work [6], the social and economic implications of this future of unpaid work could be significant. It could free up additional hours from people's lives for paid work and leisure, especially for women. It could in principle reduce the demand for domestic and care workers in aging societies like the UK and especially Japan and conversely diminish opportunities for migrant workers from other lower-income countries. However, policy makers should be mindful of the fact that the potentials appear significantly stronger in housework than in care work, especially adult care. Areas where experts saw the greatest potential for further time savings were shopping, service use, and cleaning related tasks.

That said, our sociological approach to the expert panel method used in Frey and Osborne [1,8] and its follow-up studies also highlighted an epistemological flaw in the previous predictions. By asking how technological transformation could impact on workplaces and homes rather than the other way round, these studies rely implicitly on a deterministic theory of technological progress in which technology is assumed to develop autonomously from the rest of society [40,55]. By giving a voice to the experts who are asked to make the predictions, we revealed how the experts struggled with this theorization, because their experience had taught them important ways in which the two are mutually constituted. Economic factors, social norms, values, and gendered expectations around domestic work shaped what avenues of technological development got pursued in the first place.

Indeed, scholars in social studies of science and technology have long highlighted the social shaping of technological development. Ruth Schwartz Cowan [39] showed how the adoption new technologies at home had long been shaped among other things by who controlled the household budget and whose time was valued more. More recent feminist scholarship furthermore argues that the marginalisation of women in the development of technologies has steered technologies available in the market towards directions that do not always serve women's interests [56,57]. One recent example of this may be how the kind of digitalized home schooling implemented in the UK during the pandemic meant that many carers, especially women, had to double as unpaid teaching assistants and IT support persons on top of their normal duties—with detrimental consequences to their ability to succeed in paid work careers [58]. In another telling example, female, but not male, income is positively associated with dishwasher adoption in Japan; no such difference is observed in the adoption of air conditioners, PCs, or cellular phones [59].

None of this means that "future of work" predictions are necessarily futile; it does mean, however, that they are never merely technical predictions but socio-technical forecasts—forecasts of how both technology *and* society will evolve. This has important implications to the nature of the expertise required to produce them. A group of machine learning researchers [1,8] may be relatively well placed to assess the technical feasibility of automating a task, but unqualified to assess whether social and economic pressures will result in technologists being asked to attempt to do so in the first place. In this study, we sketched out a more sociologically informed approach to forecasting and began to draw on a broader set of "AI" expertise.

We found that participants from different professional communities held somewhat different views of the automatability of unpaid work tasks, approaching as they did the socio-

technical system of domestic work from different directions. We also found that our UK experts were on average more optimistic about the prospects of domestic automation than our Japanese experts were. Some of the differences were quite notable in magnitude; for instance, UK experts' 10-year estimates were on average 33 percent higher than Japanese experts' estimates. These differences remained even after a Delphi exercise in which the participants were explicitly invited to learn from each others' diverse perspectives.

Underlying the country difference in our sample we also detected a strong gender dimension: our UK male experts were notably optimistic about the prospects of domestic automation, while our Japanese male experts were notably pessimistic, bucking usual male optimism about technology [53]. As a possible interpretation for the Japanese male experts' relative lack of enthusiasm, we pointed to the politics of the average Japanese household, where domestic work remains very much a woman's occupation. Previous studies have treated all experts as interchangeable and have not disclosed the experts' backgrounds.

Our number of respondents was large for a Delphi study [46], but like other expert studies, too small and non-probabilistic to statistically generalize the findings to all AI experts, which is an important limitation of this research. The differences that we explored could also be confounded by factors such as age. Instead, the results serve to demonstrate how the outcomes of a typical "future of" style expert study can be shaped by factors such as diverging national discourses and diverse gendered realities. They illustrate how expert opinion even on technological issues and perhaps especially on technological issues remains socially contingent; expert opinion cannot be simply taken as a source of objective "ground truth", even when it is aggregated and processed through a group forecasting method such as Delphi.

Perhaps most importantly, a socio-technical approach to forecasting asks researchers to recognize that the practice of forecasting is itself also implicated in the shaping of the future. Regardless of how accurate Frey and Osborne's [1,8] striking headline figure ultimately turns out to be [3], it has already had an impact on how policy makers, academics, investors, and the public allocate their attention and resources [60]. In other words, predictions don't just anticipate the future—they shape it [61]. Forecasts based on narrow sets of expertise are liable to generate future visions that prioritize only certain social groups' interests. To combat this, studies should seek to draw on diverse expertise and allow the findings to be contextualized by being transparent about the experts' backgrounds. Studies should also seek to broaden future visions from well-trodden paths such as employment to underexplored areas such as domestic work.

## Supporting information

**S1 Data.**
(CSV)

**S1 File.**
(DTA)

**S2 File.**
(TEXT)

**S3 File.**
(DOCX)

## Acknowledgments

We are grateful to the 65 experts who participated in our study for their generosity with their time and knowledge, without which this research would not have been possible. We are also

grateful to Jens Beckert and other organizers of the Imagined Capitalist Transformations mini-conference at the Society for the Advancement of Socio-Economics (SASE) annual conference in July 2021, and in particular to the discussant Kathleen Griesbach for her valuable feedback. We are likewise grateful to Michael Osborne and Fabian Stephany for their useful comments on earlier versions of this study.

## Author Contributions

**Conceptualization:** Vili Lehdonvirta, Lulu P. Shi.

**Data curation:** Vili Lehdonvirta, Lulu P. Shi, Ekaterina Hertog, Nobuko Nagase, Yuji Ohta.

**Funding acquisition:** Ekaterina Hertog, Nobuko Nagase.

**Methodology:** Vili Lehdonvirta, Lulu P. Shi, Ekaterina Hertog, Nobuko Nagase, Yuji Ohta.

**Writing – original draft:** Vili Lehdonvirta, Lulu P. Shi.

**Writing – review & editing:** Vili Lehdonvirta, Lulu P. Shi, Ekaterina Hertog, Nobuko Nagase, Yuji Ohta.

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
