## [Decision Letter · Decision Letter 0]

2 Nov 2022

PONE-D-22-21484The future(s) of unpaid work: How susceptible do experts from different backgrounds think the domestic sphere is to automation?PLOS ONE

Dear Dr. Lulu P Shi,

Thank you for submitting your manuscript to PLOS ONE. After careful consideration, we feel that it has merit but does not fully meet PLOS ONE’s publication criteria as it currently stands. Therefore, we invite you to submit a revised version of the manuscript that addresses the points raised during the review process.

Kindly look into comments as suggested by the reviewers for your revision.

We look forward to receiving your revised manuscript.

Kind regards,

Mahmud Iwan Solihin

Academic Editor

PLOS ONE

Journal Requirements:

“This research was supported by a UK-Japan collaborative grant jointly awarded by UK Research and Innovation (grant number ES/T007265/1; PI Ekaterina Hertog) and by the Research Institute of Science and Technology for Society (RISTEX) of the Japan Science and Technology Agency  (grant number JPMJRX19H4; PI Nobuko Nagase). This project also benefited from funding from the European Union’s Horizon 2020 research and innovation programme under the European Research Council Consolidator Grant agreement No 771736.”

“This research was supported by a UK-Japan collaborative grant jointly awarded by UK Research and Innovation (grant number ES/T007265/1; PI E.H.) and by the Research Institute of Science and Technology for Society of the Japan Science and Technology Agency (grant number JPMJRX19H4; PI N.N.). The funders did not play any role in the study design, data collection and analysis, decision to publish, or preparation of the manuscript.”

Additional Editor Comments:

Dear Authors,

I am glad to inform you that you paper (The future(s) of unpaid work: How susceptible do experts from different backgrounds think the domestic sphere is to automation?)

has been reviewed and our final decision is minor revision as suggested by our reviewers.

Congratulations and do necessary correction as suggested by the reviewers.

Reviewers' comments:

Reviewer's Responses to Questions

**Comments to the Author**

1. Is the manuscript technically sound, and do the data support the conclusions?

Reviewer #1: Yes

Reviewer #2: Yes

2. Has the statistical analysis been performed appropriately and rigorously? 

Reviewer #1: Yes

Reviewer #2: Yes

3. Have the authors made all data underlying the findings in their manuscript fully available?

Reviewer #1: Yes

Reviewer #2: Yes

4. Is the manuscript presented in an intelligible fashion and written in standard English?

Reviewer #1: Yes

Reviewer #2: Yes

5. Review Comments to the Author

Reviewer #1: Comments

1. It really is a pleasure to read this paper. I think that the authors provide a very well-crafted and well-written paper, as well as an excellent and sound analysis of the automatability of unpaid domestic work. Previous studies focused on paid work, and rarely so on paid domestic work, so the authors’ study carries novel and important research results focusing on unpaid domestic work. The authors trace out how their work differs from other works in the literature, especially in terms of the methodology, and they put emphasis on analysing the link between automation and the future of work with methods from the science of Sociology. Importantly, from interviewing experts they can reveal how differences in experts’ backgrounds, i.e. between males and females, and between the UK and Japan, shape experts’ responses towards the automatability of domestic tasks. And they emphasize that this is something that is missed out in – especially other sciences’ – datasets of labelling or expert information. The authors find, among others, that the experts on average predicted that 39 percent of domestic tasks may be automatable within the next ten years. The results carry important implications for politics, the economy and society.

2. A few more comments: I think you should write into your Abstract that you apply a sociological approach to the study of automatability of domestic work. Please do the same to the first paragraph in your last chapter (Discussion and Conclusions).

3. On page 2, in the first paragraph of the Introduction, it would be good to extend the named literature with some further studies that replicated or approached the study by Frey and Osborne. Moreover, there exists some literature on automation and paid domestic work, which could be considered as well (e.g. Krenz and Strulik 2022, Mazzolari and Ragusa 2013).

4. On pages 12, 13 and the following, you are writing about the Delphi method, which has been applied. You used the method rigorously, but please provide some further explanations on the Delphi method, its usage, and some more details for a wider audience. Also provide literature on its theory as well as some empirical applications.

5. On page 12 in line 7 an “of” is missing.

6. On page 17 in the first paragraph, please insert the relevant authors’ names for “XXX”.

7. I would suggest not to name it “Frey and Osborne (2013, 2017) style methods”, but to directly name the methodologies that you refer to, e.g. labelling, which algorithm you refer to etc., on page 21, 22 and on some previous pages, and after having named the relevant content, then cite the authors/ studies, e.g. “following the approach by Frey and Osborne” or “similar to the study by…” or “(literature, year)”.

8. On page 21, in the second paragraph, you say “Neither Frey and Osborne’s prediction nor our figure are presented as forecasts”, however on page 11, 12 and the following, as well as on page 23, you are speaking about doing “forecasting rather than labelling” and “most widely used group forecasting method today is the Delphi method”. Please make sure to rewrite the relevant passages somewhat, in order to guarantee consistency of reading.

Reviewer #2: In this manuscript, the authors conducted a forecasting exercise in which 65 AI experts from the UK and Japan estimated how automatable are 17 housework and care work tasks.

Comments and Suggestions for Authors

- The authors used 285 experts based in the UK and Japan. They tried to identify comparable numbers of AI experts working in three different professional settings: academia, R&D (understood here as corporate research labs and start-ups), and business (e.g., venture capital and marketing). Any people works in automation/robotics companies are involved in this research? It will be very useful if we could get information from the manufacturer regarding their opinions, strategies, and plan to support the automation technology in the field of unpaid work.

- Instead of only discussing the impact on two modern countries, ie. Japanese and UK, could you explain more on how this research will impact the rest of the world? The automation is involve in the world, not only for the first world countries but also in third middle countries which are usually known as labor supply for unpaid worker. Possibly the motivation could be extend also to more general instead of only UK and Japan based analysis.

- Page 17, In the same vein XXX’s study suggest that >> are you sure this is XXX’s study?

6. PLOS authors have the option to publish the peer review history of their article (what does this mean?). If published, this will include your full peer review and any attached files.

Reviewer #1: No

Reviewer #2: No

---

## [Author Response · Author response to Decision Letter 0]

30 Dec 2022

Dear editor,

1. Thank you very much for your useful comments on our manuscript, titled "The future(s) of unpaid work: How susceptible do experts from different backgrounds think the domestic sphere is to automation". We have revised the manuscript accordingly. As requested, please find attached the revised manuscript with changes tracked, the revised manuscript without tracked changes, and our response letter to the reviewers.

2. Regarding the funding statement, we have removed the funding related information from the manuscript as requested. Please could you use the following as our funding statement:

"This research was supported by a UK-Japan collaborative grant jointly awarded by UK Research and Innovation (grant number ES/T007265/1; PI Ekaterina Hertog) and by the Research Institute of Science and Technology for Society (RISTEX) of the Japan Science and Technology Agency (grant number JPMJRX19H4; PI Nobuko Nagase). This project also benefited from funding from the European Research Council (ERC) under the European Union's Horizon 2020 research and innovation programme under grant agreement No 681546 (FAMSIZEMATTERS) and grant agreement No 771736 (GENTIME). The funders did not play any role in the study design, data collection and analysis, decision to publish, or preparation of the manuscript."

3. The ethics statement information is fully included in the methods section of the manuscript.

4. Regarding data availability, we have now submitted the data to the UK Data Service, which is a national public repository. However, it takes some time for the submission to be reviewed and published, so we do not have a DOI for the data at this moment. In the meantime, we are uploading the data as a Supporting Information. We hope to update you with the DOI as soon as we have it.

Sincerely,

Vili Lehdonvirta, Lulu P. Shi, Ekaterina Hertog

***

24 November 2022

Response to reviewers

PONE-D-22-21484 The future(s) of unpaid work: How susceptible do experts from different backgrounds think the domestic sphere is to automation?

Dear reviewers,

Thank you very much for your constructive feedback on our manuscript. We feel that addressing your comments has further strengthened the manuscript. Please see below our responses (bold) to your comments (normal) and a short description of how we have addressed each comment in the manuscript.

Kind regards,

The authors

Reviewer #1: Comments

1. It really is a pleasure to read this paper. I think that the authors provide a very well-crafted and well-written paper, as well as an excellent and sound analysis of the automatability of unpaid domestic work. Previous studies focused on paid work, and rarely so on paid domestic work, so the authors’ study carries novel and important research results focusing on unpaid domestic work. The authors trace out how their work differs from other works in the literature, especially in terms of the methodology, and they put emphasis on analysing the link between automation and the future of work with methods from the science of Sociology. Importantly, from interviewing experts they can reveal how differences in experts’ backgrounds, i.e. between males and females, and between the UK and Japan, shape experts’ responses towards the automatability of domestic tasks. And they emphasize that this is something that is missed out in – especially other sciences’ – datasets of labelling or expert information. The authors find, among others, that the experts on average predicted that 39 percent of domestic tasks may be automatable within the next ten years. The results carry important implications for politics, the economy and society.

Thank you very much for these encouraging words and for clearly identifying our contribution. We really appreciate it.

2. A few more comments: I think you should write into your Abstract that you apply a sociological approach to the study of automatability of domestic work. Please do the same to the first paragraph in your last chapter (Discussion and Conclusions).

Thank you for this excellent suggestion. We added “sociological approach” to the first paragraph of the introduction and to the fourth paragraph of the Discussion and Conclusions section, where it seemed to fit better.

3. On page 2, in the first paragraph of the Introduction, it would be good to extend the named literature with some further studies that replicated or approached the study by Frey and Osborne. Moreover, there exists some literature on automation and paid domestic work, which could be considered as well (e.g. Krenz and Strulik 2022, Mazzolari and Ragusa 2013).

Thank you for these great suggestions. We added two citations to the introductory paragraph and cited Krenz and Strulik (2022) in the literature review as an example of a study of the impacts of automation on paid domestic work.

4. On pages 12, 13 and the following, you are writing about the Delphi method, which has been applied. You used the method rigorously, but please provide some further explanations on the Delphi method, its usage, and some more details for a wider audience. Also provide literature on its theory as well as some empirical applications.

Thank you for this suggestion. We expanded the paragraph introducing the Delphi method with slightly more elaboration on the theory and examples of applications, plus additional literature.

5. On page 12 in line 7 an “of” is missing.

Fixed, thank you.

6. On page 17 in the first paragraph, please insert the relevant authors’ names for “XXX”.

Apologies, corrected!

7. I would suggest not to name it “Frey and Osborne (2013, 2017) style methods”, but to directly name the methodologies that you refer to, e.g. labelling, which algorithm you refer to etc., on page 21, 22 and on some previous pages, and after having named the relevant content, then cite the authors/ studies, e.g. “following the approach by Frey and Osborne” or “similar to the study by…” or “(literature, year)”.

Thank you for pointing out. We added some clarifying language, such as “expert panel” to pages 21 and 22.

8. On page 21, in the second paragraph, you say “Neither Frey and Osborne’s prediction nor our figure are presented as forecasts”, however on page 11, 12 and the following, as well as on page 23, you are speaking about doing “forecasting rather than labelling” and “most widely used group forecasting method today is the Delphi method”. Please make sure to rewrite the relevant passages somewhat, in order to guarantee consistency of reading.

Thank you for pointing out this seeming contradiction. We rephrased the passage on p. 21 to make our intention clear: “Neither Frey and Osborne’s prediction nor our figure are intended as prognoses of what will inevitably happen, but rather as estimates of where the technical frontiers lie-”

Reviewer #2: In this manuscript, the authors conducted a forecasting exercise in which 65 AI experts from the UK and Japan estimated how automatable are 17 housework and care work tasks.

Comments and Suggestions for Authors

- The authors used 285 experts based in the UK and Japan. They tried to identify comparable numbers of AI experts working in three different professional settings: academia, R&D (understood here as corporate research labs and start-ups), and business (e.g., venture capital and marketing). Any people works in automation/robotics companies are involved in this research? It will be very useful if we could get information from the manufacturer regarding their opinions, strategies, and plan to support the automation technology in the field of unpaid work.

Thank you for this comment and question. Yes, many or most of the companies included in the sample deal with automation/robotics in some way. But the sampling criteria were based on the expertise of the individuals rather than on the companies, so we did not collect systematic data on this. We have added more detail on the sampling process to the manuscript.

- Instead of only discussing the impact on two modern countries, ie. Japanese and UK, could you explain more on how this research will impact the rest of the world? The automation is involve in the world, not only for the first world countries but also in third middle countries which are usually known as labor supply for unpaid worker. Possibly the motivation could be extend also to more general instead of only UK and Japan based analysis.

Thank you for this excellent suggestion. We added a little bit of elaboration on possible effects on different countries to the discussion section. In future research we aim to analyse this issue more extensively.

- Page 17, In the same vein XXX’s study suggest that >> are you sure this is XXX’s study?

Apologies, corrected!

---

## [Editor Report · Decision Letter 1]

20 Jan 2023

The future(s) of unpaid work: How susceptible do experts from different backgrounds think the domestic sphere is to automation?

PONE-D-22-21484R1

Dear Dr. Shi,

We’re pleased to inform you that your manuscript has been judged scientifically suitable for publication and will be formally accepted for publication once it meets all outstanding technical requirements.

Kind regards,

Mahmud Iwan Solihin

Academic Editor

PLOS ONE
---

## [Editor Report · Acceptance letter]

30 Jan 2023

PONE-D-22-21484R1 

The future(s) of unpaid work: How susceptible do experts from different backgrounds think the domestic sphere is to automation? 

Dear Dr. Shi:

I'm pleased to inform you that your manuscript has been deemed suitable for publication in PLOS ONE. Congratulations! Your manuscript is now with our production department. 

Kind regards, 

on behalf of

Dr. Mahmud Iwan Solihin 

Academic Editor

PLOS ONE